# Usefulness of Tree Species as Urban Health Indicators

**DOI:** 10.3390/plants10122797

**Published:** 2021-12-17

**Authors:** Edina Simon, Vanda Éva Molnár, Domonkos Lajtos, Dina Bibi, Béla Tóthmérész, Szilárd Szabó

**Affiliations:** 1Department of Ecology, Faculty of Science and Technology, University of Debrecen, H-4032 Debrecen, Hungary; lajtos.domo@gmail.com (D.L.); dinaangel172@gmail.com (D.B.); 2Department of Physical Geography and Geoinformatics, Faculty of Science and Technology, University of Debrecen, H-4032 Debrecen, Hungary; molnarvandaeva@science.unideb.hu (V.É.M.); szabo.szilard@science.unideb.hu (S.S.); 3MTA-DE Biodiversity and Ecosystem Services Research Group, H-4032 Debrecen, Hungary; tothmerb@gmail.com

**Keywords:** urbanisation, metals, plants, air pollution, ICP-OES, environmental health

## Abstract

We used the Air Pollution Tolerance Index (APTI), the amount of PM_5_ and PM_10_, and the elemental analysis of leaves to explore the sensitivity of tree species to air pollution. We assessed the tolerance of *Robinia pseudoacacia*, *Acer saccharinum*, *Tilia* × *europaea*, *Acer platanoides*, *Fraxinus excelsior*, *Betula pendula*, *Celtis occidentalis*, and *Platanus* × *acerifolia* to the amount of dust, APTI, and the elemental concentration of leaves. Leaves were collected in Debrecen (Hungary), which has a high intensity of vehicular traffic. The highest amount of PM (both PM_10_ and PM_5_) was found on the leaves of *A*. *saccharinum* and *B*. *pendula*. Our results demonstrated that *A*. *saccharinum* was moderately tolerant, while *P*. *acerifolia* was intermediate, based on the APTI value. There was a significant difference in the parameters of APTI and the elemental concentration of leaves among species. We found that tree leaves are reliable bioindicators of air pollution in urban areas. Based on the value of APTI, *A*. *saccharinum* and *P*. *acerifolia*, and based on PM, *A*. *saccharinum* and *B*. *pendula* are recommended as pollutant-accumulator species, while other studied species with lower APTI values are useful bioindicators of air pollution. The results support landscape engineers and urban developers in finding the best tree species that are tolerant to pollution and in using those as proxies of urban environmental health.

## 1. Introduction

Air pollution is an increasing problem worldwide. While in Europe air pollution decreases, it is still a problem, especially in cities; the urban and industrial areas are diverse sources of various pollutants that are a health risk for the human population [1,2]. Plants and the soil are primary sinks for air and soil pollution; the heavy metals translocated via atmospheric deposition negatively impact ecosystems [3]. Moreover, urban green spaces are vital in protecting biodiversity and improving the quality of urban life [4,5].

The accumulation of contaminants by plants is widely documented [6,7,8,9,10]. Trees are particularly efficient in trapping dust-reducing airborne particles deposited in tree leaves’ stomatal openings and waxy cuticles [11]. Leaves accumulate metals and other pollutants in high quantities, which is higher than the maximum limit based on WHO/FAO recommendations for vegetables [12,13]. Naturally, metals can also accumulate via the root system, but the intensity of translocation through plant parts depends on the given metal and plant species. Plants’ susceptibility and tolerance varies by species as a function of air pollution types, such as high concentration of carbon monoxide, nitrogen dioxide, sulphur dioxide, and particulate matter [14].

Atmospheric particulate matter (PM) consists of a complex mixture of substances and chemical compounds of natural and/or anthropogenic origin [15]. Generally, PMs are framed in terms of two health-relevant fractions, from which the first is referred to as PM_2.5_ and the other as PM_10_ [16]. PM is currently considered to be the best indicator for the health effects of ambient air pollution [17] because it is a typical pollutant in urban areas due to its ability to bind toxic organic components and heavy metals [18,19]. Accordingly, PM pollution causes serious health issues for the urban population [19,20]. Composite particulate matter is directly linked to asthma, eye and lung problems, and premature death. It can be emitted directly into the air and can be formed in the atmosphere. These particles are multicomponent aggregates comprising a wide range of substances [2]. PM_10_ and coarse particles refer to PM with an aerodynamic diameter of <10 µm, and fine particles refer to PM with a diameter of <2.5 μm, while ultrafine particles are those with a diameter of <0.1 um [21]. Small particles of 3.3–4.7 µm diameter can penetrate the trachea and primary bronchi; thus, studying PM_5_ is an important task in urbanised areas [21,22,23,24]. As a result of human activities, the background level of heavy metals is increasing and accumulates in the soil and vegetation due to their non-biodegradable nature [25]. Plants, especially trees and shrubs, play an important role in the adsorption and reduction of PM concentrations in the air because they are effective at accumulating PM on the surface of leaves, stems, and bark [26,27]. The physiological state of leaves provides a simple, cheap, but still useful indicator for biomonitoring [28,29,30]. An efficient indicator, APTI, expresses the ability of plants to counter the adverse effects of air pollution [28]. APTI was proposed by Singh and Rao [28] using the ascorbic acid content, total chlorophyll content, pH of leaf extract, and relative water content of leaves to quantify the reaction of plants to air pollution. Thus, the level of air pollution can be expressed by the index as an indirect reaction of plants. Furthermore, the index also made it possible to compare these reactions and to identify species with less sensitivity that can be optimally used in urban green infrastructure planning; high values of APTI indicate low sensitivity, while tree species of low APTI can be considered biological pollution indicators [6,7,28,29,30].

An earlier study demonstrated that correlations between the amount and elemental concentrations of dust deposited on tree leaves, and APTI values confirmed that APTI efficiently indicated the level of air pollution [29]. The tolerance and sensitivity vary among species. Accordingly, we aimed to analyse the sensitivity of eight frequent European tree species to air pollution based on the PM amount, APTI, and elemental concentration (Al, Ba, Ca, Cu, Fe, K, Li, Mg, Mn, Na, Ni, P, Pb, S, Sr, and Zn) of tree leaves. 

## 2. Results

### 2.1. Particulate Material (PM) and Air Pollution Tolerance Index (APTI) 

There were significant differences in the amount of PM_5_ and PM_10_ among the species (PM_5_: F = 247.312, *p* < 0.001; PM_10_: F = 48.937, *p* < 0.001). The highest dust amounts were found on the leaves of *A*. *saccharinum* and *B*. *pendula* for both PM_10_ and PM_5_ (Figure 1).

According to the categorisation of Singh et al. [26], most trees were sensitive, except *A*. *saccharinum* and *P*. *acerifolia*. *A*. *saccharinum* was moderately tolerant, while *P*. *acerifolia* was intermediate (Figure 2).

Differences in the means of parameters of APTI values (ascorbic acid, total chlorophyll content, pH of leaf extract, and relative water content) were tested among the species. There were significant differences among species based on leaves for the ascorbic acid content (F = 48.006, *p* < 0.001), for the pH of the leaf (F = 48.669, *p* < 0.001), and for the total chlorophyll content of leaves (F = 4.662, *p* = 0.006) (Figure 3a–c). There were no significant differences among species based on the relative water content of leaves (F = 2.250, *p* = 0.089) (Figure 3d).

The ascorbic acid content was the highest in *B*. *pendula* leaves, and the pH was highest in the dust adsorbed in the leaves of *R*. *pseudoacacia* and *C*. *occidentalis*. The total chlorophyll content was highest in the leaves of *A*. *platanoides*. 

### 2.2. Elemental Concentration of Leaves 

The first two Discriminant Functions of Canonical Discriminant Analysis had more than 10% explained variance (DF1 = 80.5%, DF2 = 10.2%) and provided a cumulative variance of 90.7%. Furthermore, canonical correlations were about 0.98 for these DFs, but, except for DF6 and DF7, correlations were >0.9 (Figure 4, Appendix A). 

DF2 positively correlated with Ca (r = 0.483), Sr (r = 0.263), and Ba (r = 0.160), while we found a negative correlation with the *p* concentration and DF3 (r = −0.201). There was a negative correlation between DF4 and the Li concentration (r =− 0.322), while Zn (r = −0.571), K (r = −0.333), and Mg (r = 0.281) correlated with DF5. DF6 negatively correlated with S (r = −0.455), Ni (r = −0.338), and K (r = −0.150), while a positive correlation was found between Al (r = 0.594) and Pb (r = 0.312). There was a negative correlation between DF7 and Cu (r = −0.395) and Fe (r = −0.337), while Mn (r = 0.385) was positively correlated. 

There were significant differences in the Al (F = 17.572, *p* < 0.001), Ba (F = 4.251, *p* = 0.009), Ca (F = 24.675, *p* < 0.001), Fe (F = 10.856, *p* < 0.001), Mg (F = 2.777, *p* = 0.046), Ni (F = 4.715, *p* = 0.006), S (F = 7.119, *p* = 0.001), Sr (F = 8.097, *p* < 0.001), and Zn (F = 5.353, *p* = 0.003) concentrations in leaves among the species (Table 1). 

The highest Al and Fe concentrations were found in the leaves of *T*. × *europaea* and *A*. *platanoides*. The Ba concentration was the highest in the leaves of *R*. *pseudoacacia* and *C*. *occidentalis*, similar to the Ca concentration. For Mg and Ni, *P*. *acerifolia* had the highest concentration. The highest S concentration was observed in *C*. *occidentalis*. The Sr concentration was the highest in *A*. *platanoides*, and the Zn concentration was the highest in *F*. *excelsior* leaves (Table 1).

## 3. Discussion

We demonstrated that urban tree leaves can be used effectively for monitoring urban air quality. Leaves act as dust and air pollutant traps due to specific factors in the anatomy of tissues, such as trichomes and stomata density. There were significant differences in the amounts of PM_5_ and PM_10_ among the species. The highest dust amount was found on the leaves of *A*. *saccharinum* and *B*. *pendula* for both PM_10_ and PM_5_. We demonstrated that most tree species were sensitive, except for *A*. *saccharinum* and *P*. *acerifolia*. *A*. *saccharinum* was moderately tolerant, while *P*. *acerifolia* was intermediate, based on the APTI value. There was a significant difference among species based on leaves for the ascorbic acid content, for the pH of the leaf, and for the total chlorophyll content of leaves. The ascorbic acid content was the highest in *B*. *pendula* leaves, and the pH was highest in the leaves of *R*. *pseudoacacia* and *C*. *occidentalis*. The total chlorophyll content was highest in the leaves of *A*. *platanoides*. There was a significant difference in the Al, Ba, Ca, Fe, Mg, Ni, S, Sr, and Zn concentrations in leaves among the species.

An increase in PM_10_ concentration was revealed in Lithuania and Romania, while a slighter reduction in PM_10_ emissions in Bulgaria and Hungary from 2000–2017 was observed by Sicard et al. [1]. Despite an earlier study [30], our results indicated that the leaves of *A*. *saccharinum* and *B*. *pendula* were useful for assessing the level of air pollution for PM_10_ and PM_5_. Kretinin and Selyanina [30] reported that *T*. × *europaea* had a very high dust retaining capacity (464 g m^−2^), *F*. *excelsior* had a moderate capacity (41.5 g m^−2^), while *A*. *saccharinum* had a reduced capacity (13.9 g m^−2^), and *B*. *pendula* had a low dust-retaining capacity (4.8 g m^−2^). The most advantageous characteristics are rough and tomentose tree leaves, which result in a very high dust-retaining capacity; thin waxy leaves ensure a high and moderate dust-retaining capacity, allowing dust particles to stick to the leaf surface. However, tree leaves with dense glossy surfaces are enough only for reduced and low dust-retaining capacity [30]. *A*. *saccharinum* is a tree species of great capacity to remove PM_2.5_ effectively from urban air [31]. *P*. *acerifolia* was considered a less sensitive species to SO_2_ and fluorides, and very tolerant to pollution [32]. Haynes et al. [33] studied the effect of urbanisation on the amount of PM using moss turf and a common native tree species with *Pittosporum undulatum*. Their results indicated that moss species are more sensitive to increasing urbanisation, based on the amount of PM, than trees. At the same time, the amount of PM on the leaf’s surface of *P*. *undulatum* was similar to our results for *F*. *excelsior* and *T*. *× europaea*. Similar to our findings in an earlier study on the foliage of *B*. *pendula*, its leaves collected the most PM, followed by *Q*. *robur* and *T*. *cordata*, regardless of the dust’s source [26]. Łukowski et al. [26] demonstrated that *B*. *pendula*, *Q*. *robur*, and *T*. *cordata* had a tendency for higher wax production when grown under higher PM pollution levels. The amount of PM was similar to our findings for *B*. *pendula* and *T*. *cordata*, which were reported by Łukowski et al. [26]. Popek et al. [27] also studied the difference between native and non-native tree species based on PM accumulation. Their results showed that both *P*. *padus* and *P*. *serotina* accumulated the most PM on the surface of the leaves rather than in the wax layer. Furthermore, the native *P*. *padus* accumulated higher amounts of PM than *P*. *serotina*. The results about the surface of leaves and the wax layer of leaves indicated that they play the most important role in the accumulation of PM. Based on the APTI values, we demonstrated that *A*. *saccharinum* was moderately tolerant, while *P*. *acerifolia* was intermediate. Alotaibi et al. [34] reported that the reduction of leaf area was significantly recorded in contaminated areas for *P*. *acerifolia*. These findings suggest that the differences in the reduction in leaf areas between tree species at different locations may be due to the availability of leaf surface area and the capacity of leaves for capturing air pollutants [35]. Our results were similar to Chen et al. [36], in which *P*. *acerifolia* was tolerant to pollution and should be given preference in plant selection for seriously polluted places. Nadgórska-Sochaet al. [37] demonstrated that based on APTI and the elemental concentration, *B*. *pendula* and *Taraxacum officinale* may be appropriate plants in urban areas with considerable soil and air contamination, especially with heavy metals. The value of APTI for *B*. *pendula* and *R*. *pseudoacacia* was higher in their study than in our results. A similar result was found by Rai et al. [38], who demonstrated that the tolerance of plants towards air pollution may be site-specific because *Ficus bengalensis* was found to be tolerant in industrial sites and *Mangifera indica* was tolerant in non-industrial sites. Ogunkunle et al. [39] also suggested that the integration of both plant tolerance and performance indices for the selection of tree species are very useful for the development of a green belt using APTI values. Their results demonstrated that based on APTI, *Vitellaria paradoxa*, *Terminalia catappa*, *Acacia nilotica*, and *Prosopis africana* are sensitive species to air pollution stress in Nigeria. Bharti et al. analysed the APTI values of 25 plant species, and *Ficus bengalensis*, *Ficus religiosa*, *Eucalyptus globus*, *Azadirachta indica juss*, and *Heveabra brasiliensis* were tolerant to air pollution, while *Polythalia longifolia* was found to be the most sensitive. They also demonstrated that species with an APTI < 11 may be used as a bioindicator of air quality, while those with an APTI ≥ 17 can be used for green belt design. Jyothi and Jaya [40] also reported that tolerant tree species can serve as sinks, and sensitive tree species can act as indicators for air pollution mitigation. Gholami et al. [41] also found that plants with higher APTI can be used as reducers of pollution, and plants with a lower APTI can be used to measure air pollution.

## 4. Materials and Methods

### 4.1. Study Sites and Sample Collection

The sampling area was located in Debrecen (second largest city of Hungary), near the city centre, which is exposed to a high intensity of vehicular traffic (Figure 5). From 2000–2017, the average concentration of PM_10_ was lower (36 ± 4 µg m^−3^) and PM_2.5_ was higher (28 ± 3 µg m^−3^) than the threshold limit in Directive 2008/50/EC of the European Parliament (PM_10_ = 40 µg m^−3^ and PM_2.5_ = 25 µg m^−3^) [42,43]. We tested the usefulness of the following species as bioindicators of air pollution: *R*. *pseudoacacia* (Linné)., *A*. *saccharinum* (Linné), *T*. × *europaea* (Linné), *A*. *platanoides* (Linné), *F*. *excelsior* (Linné), *B*. *pendula* (Roth), *C*. *occidentalis* (Linné), and *P*. *acerifolia* (Münchhausen). These species were the most dominant in the studied area. The age of individuals was about 10–15 years, the state of trees was healthy, and the presence of pests and symptoms of disease were not found by visual examination. We randomly chose 3 individuals from each species and collected 15 leaves from each tree at a 1.5 m height.

### 4.2. Dust Amount Analysis

The area of the sampled leaf was determined by scanning the leaves in black and white, and the pictures were analysed using ImageJ software. During chemical analyses, the leaves were put into 500 mL plastic boxes. Then, 250 mL of deionised water was added to the leaves. The samples were shaken for 10 min on an orbital shaker. This suspension was filtered through a 150 μm sieve. Then, the leaves were again shaken in 50 mL deionised water, repeating the previous procedure. This 300 mL suspension was filtered through two types of filter paper using a vacuum filter machine (N 811 KN.18 Laboport). First, filter paper with a retention diameter of 5–8 µm was used (Munktell 392, Ahlstrom) so that the amount of coarse dust could be measured. Then, the filtrate was filtered again using a filter paper with a retention diameter of 2–3 µm (Munktell 391, Ahlstrom). The amount of fine dust was measured using the gravimetric method; filter papers were weighed before and after filtration to determine the amount of dust collected on the paper. The amount of dust was determined in µg cm^−2^, as in mass per area of the leaf’s surface [7,8,9].

### 4.3. Air Pollution Tolerance Index (APTI)

APTI values were calculated based on the ascorbic acid content in mg g^−1^ (A), total chlorophyll content in mg g^−1^ (T), pH of leaf extract (P), and relative water content (R) of the tree leaves. Using these parameters, we applied the equation proposed by Singh et al. [44] for APTI:APTI = [A × (T + P) + R]/10

The ascorbic acid content was measured with the redox titration method, where 2 g of leaf tissue was crushed and homogenised in deionised water. After filtration, the samples were titrated using an iodine solution with starch as an indicator. 

Chlorophyll was extracted from approximately 20 mg of fresh leaf tissue using 5 mL of 96% ethanol. The absorbances of the extracts were measured at wavelengths of 653, 666, and 750 nm, using spectrophotometric analysis. The total chlorophyll content (T) was calculated as follows:T (mg g^−1^) = (17.12 × E666 – 8.68 × E653) × V/m × 1000
where V is the volume (ml) of leaf extract, m is the fresh weight (g) of the leaf sample, and E666 and E653 are the absorbances at 666 nm and 653 nm minus the absorbance at 750 nm, respectively. For the pH measurement, 2 g of leaf tissue was crushed and homogenised in 100 mL deionised water. The leaf pH of this extract was measured using a digital pH meter. To determine the relative water content, the fresh weight of individual leaves (FW) was measured. Then, the leaves were immersed in water overnight before being weighed again to determine the turgid weight (TW). Finally, the leaves were dried in an oven at 70 °C to measure the dry weight (DW). The relative water content (R) was calculated as follows:R (%) = (FW – DW)/(TW – DW) × 100

Categorisation of tree species based on APTI values [45] are the following: APTI > 24, tolerant; 20–24, moderately tolerant; 15–19, intermediate; and ≤14, sensitive.

### 4.4. Elemental Analysis in Leaves

After the determination of dust amount and APTI, leaf samples were dried for 24 h at 60 °C, then the samples were homogenised and stored in plastic tubes until pre-treatment. For elemental analysis, 0.2 g of plant tissue was digested using 5 mL 65% (m/m) nitric acid and 1 mL 30% (m/m) hydrogen peroxide. Digested samples were diluted to 25 mL with deionised water [6,8,9,10]. Inductively coupled plasma optical emission spectrometry (ICP-OES 5110 Agilent Technologies) was used during the elemental analysis of leaf samples (Appendix A). We used six-point calibration procedures with a multi-element calibration solution (Merck ICP multi-element standard solution IV) and measured the concentration of Al, Ba, Ca, Cr, Cu, Fe, K, Mg, Mn, Na, Ni, Pb, and Zn. Road dust (BCR670) and Peach leaves (1547) CRM were used, and the recoveries were within 10% of the certified values for the elements. 

### 4.5. Statistical Analyses

The normality of the distribution was tested using the Shapiro–Wilk test. The homogeneity of variances was tested with Levene’s test. The differences among samples were tested using analysis of variance (ANOVA) for each variable. Tukey’s test was used for pairwise comparison between the groups. Canonical discriminant analysis (CDA) was used to reduce dimensions and to identify those variables that most efficiently discriminated the study area as the dependent variable. Dust content, total chlorophyll content, ascorbic acid content, leaf pH, and relative water content were used as independent variables. We repeated the CDA with the elemental concentrations (aluminium, barium, calcium, chromium, cobalt, copper, iron, potassium, magnesium, manganese, sodium, nickel, lead, strontium, and zinc) in leaf tissues as independent variables to separate the study area as a dependent variable. We reported the properties of discriminant functions (DFs) and their correlations with the independent variables’ observed values (r). Statistical analyses were conducted with SPSS Statistics 20 (IBM) statistical software [44].

## 5. Conclusions

We aimed to assess the role of eight tree species in urban green area planning from the aspect of pollution tolerance and bioindication of air pollution using the APTI. We demonstrated that *R*. *pseudoacacia*, *T*. × *europaea*, *A*. *platanoides*, *F*. *excelsior*, *B*. *pendula*, and *C*. *occidentalis* were sensitive indicator species of air pollution. Tolerance was moderate for *A*. *saccharinum*, while *P*. *acerifolia* was intermediate, based on the APTI value. There was a significant difference among species based on leaves for the ascorbic acid content, for the pH of the leaf, and for the total chlorophyll content of leaves. The ascorbic acid content was the highest in *B*. *pendula* leaves, and the pH was the highest on the leaves of *R*. *pseudoacacia* and *C*. *occidentalis*. The total chlorophyll content was the highest on the leaves of *A*. *platanoides*. There were significant differences in the Al, Ba, Ca, Fe, Mg, Ni, S, Sr, and Zn concentrations of leaves among the species. We found that tree leaves are reliable bioindicators of urban air pollution. APTI is useful in selecting pollution-tolerant species and can be used for urban green infrastructure planning in the phase of species selection. Based on the APTI, *A*. *saccharinum* and *P*. *acerifolia*, and based on the PM, *A*. *saccharinum* and *B*. *pendula*, are recommended as pollutant-accumulator species, while other studied species, especially those with lower APTI values, are useful bioindicators of air pollution and proxies of urban health.

## Figures and Tables

**Figure 1 plants-10-02797-f001:**
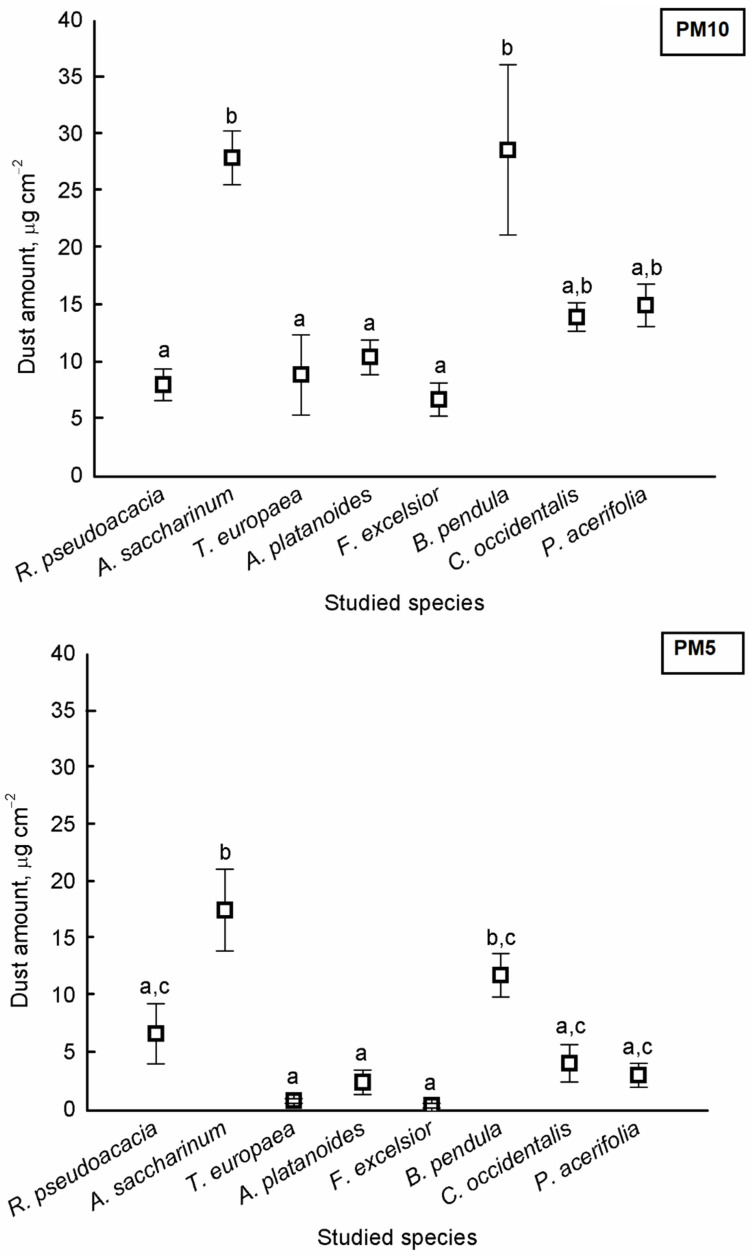
Average amount of particulate matter (±standard error) of aerodynamic diameter smaller than 10 μm (PM_10_) and smaller than 5 μm (PM_5_) on the surface of studied tree species. Different letters indicate significant differences at *p* < 0.05 level.

**Figure 2 plants-10-02797-f002:**
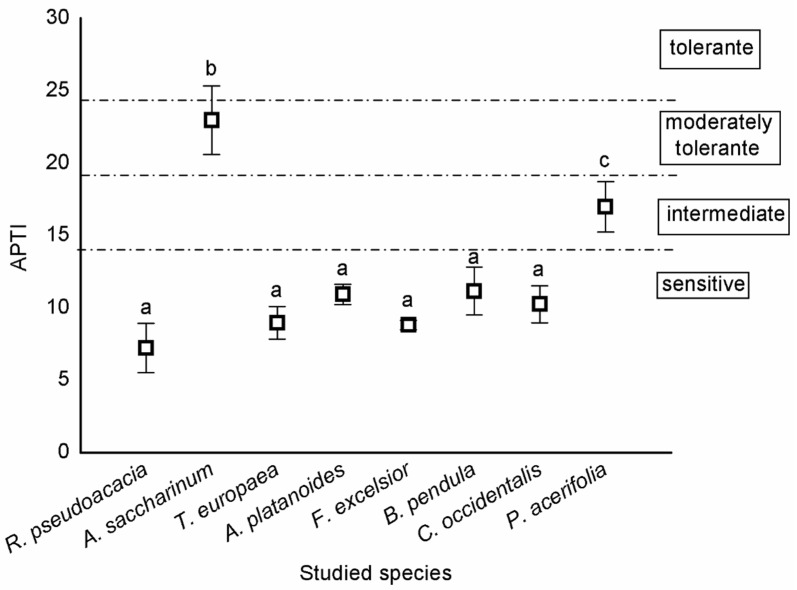
Sensitivity of tree species based on APTI values (mean ± standard error). Different letters indicate significant differences at *p* < 0.05 level.

**Figure 3 plants-10-02797-f003:**
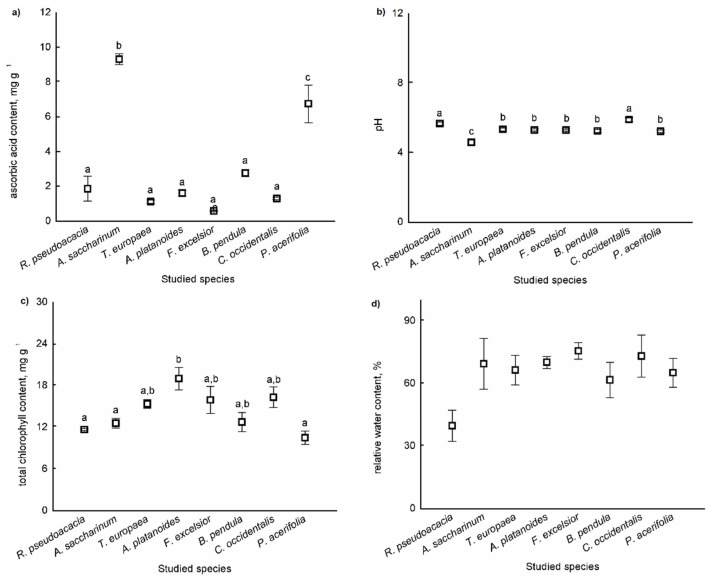
Parameters of APTI values (ascorbic acid content (**a**), pH of the leaf (**b**), total chlorophyll content (**c**), and relative water content (**d**)) of leaves (mean ± standard error). Different letters indicate significant differences at *p* < 0.05 level.

**Figure 4 plants-10-02797-f004:**
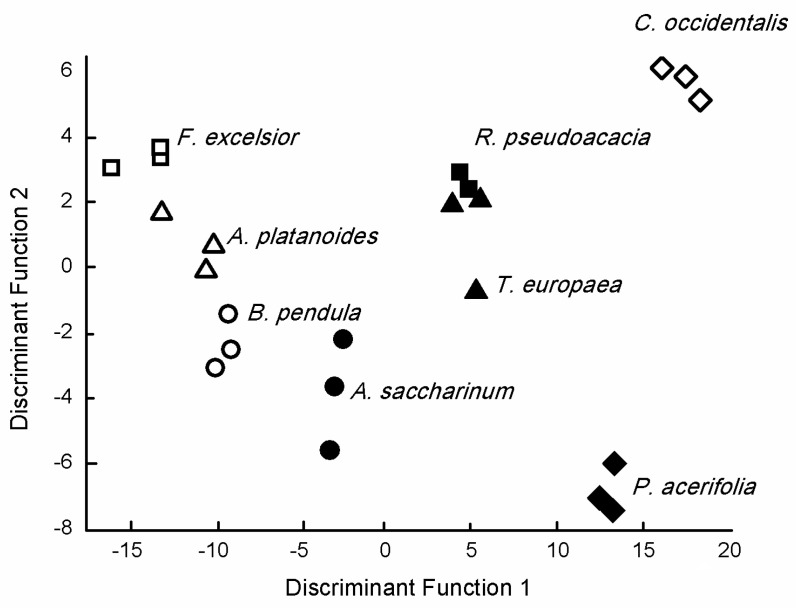
Discriminant score plots of the species based on the elemental concentration of tree species leaves.

**Figure 5 plants-10-02797-f005:**
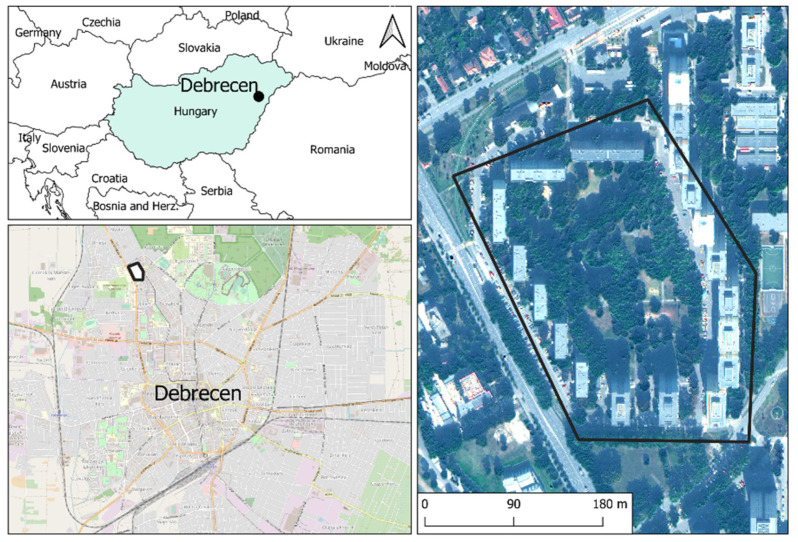
Sampling sites in Debrecen, Hungary.

**Table 1 plants-10-02797-t001:** Elemental concentration of leaves in studied species (mean ± standard error).

Elements	*R*. *pseudoacacia*	*A*. *saccharinum*	*T*. × *europaea*	*A*. *platanoides*	*F*. *excelsior*	*B*. *pendula*	*C*. *occidentalis*	*P*. *acerifolia*
Al, mg kg^−1^	12.3 ± 1.1	21.3 ± 2.6	48.7 ± 3.6	34.0 ± 4.7	15.8 ± 0.9	12.6 ± 2.3	24.6 ± 2.2	23.6 ± 2.8
Ba, mg kg^−1^	26.8 ± 8.7	5.9 ± 0.8	10.8 ± 2.1	9.6 ± 1.1	10.8 ± 3.7	14.6 ± 2.0	30.8 ± 9.3	7.2 ± 1.4
Ca, g kg^−1^	23.2 ± 1.5	12.5 ± 1.3	22.4 ± 2.8	16.9 ± 0.8	17.8 ± 4.0	12.4 ± 0.7	46.2 ± 2.9	14.1 ± 1.2
Cu, mg kg^−1^	5.1 ± 0.7	8.6 ± 1.6	6.8 ± 0.9	7.9 ± 1.7	9.5 ± 1.1	6.1 ± 0.6	6.2 ± 0.3	4.4 ± 0.2
Fe, mg kg^−1^	72.2 ± 3.1	69. 7 ± 0.7	101.2 ± 2.8	92.0 ± 10.0	62.7 ± 8.8	46.2 ± 2.8	86.4 ± 6.6	55.5 ± 1.9
K, g kg^−1^	5.9 ± 1.5	6.8 ± 0.7	4.4 ± 0.3	6.6 ± 2.9	9.0 ± 0.5	8.7 ± 0.6	4.3 ± 1.9	6.5 ± 1.4
Li, mg kg^−1^	0.4 ± 0.1	0.5 ± 0.1	0.4 ± 0.1	0.5 ± 0.1	0.3 ± 0.1	0.4 ± 0.1	0.7 ± 0.1	0.6 ± 0.1
Mg, g kg^−1^	2.5 ± 0.3	2.4 ± 0.1	1.7 ± 0.3	2.9 ± 0.1	2.2 ± 0.4	0.5 ± 0.1	2.6 ± 0.6	4.0 ± 0.5
Mn, mg kg^−1^	24.2 ± 3.8	160.1 ± 91.8	29.0 ± 4.5	41.3 ± 4.9	67.0 ± 20.3	2.8 ± 0.2	31.5 ± 9.6	27.5 ±7.4
Na, g kg^−1^	0.1 ± 0.1	0.7 ± 0.1	1.3 ± 0.6	0.7 ± 0.1	1.3 ± 0.4	36.6 ± 6.6	0.6 ± 0.1	0.6 ± 0.1
Ni, mg kg^−1^	0.4 ± 0.1	0.4 ± 0.1	1.3 ± 0.4	0.5 ± 0.2	0.6 ± 0.1	0.7 ± 0.1	1.4 ± 0.5	4.5 ± 1.7
P, g kg^−1^	1.7 ± 0.1	1.9 ± 0.3	2.0 ± 0.1	2.2 ± 0.1	2.1 ± 0.4	0.4 ± 0.1	1.3 ± 0.2	1.8 ± 0.2
Pb, mg kg^−1^	1.3 ± 0.4	1.2 ± 0.1	1.5 ± 0.2	1.7 ± 0.1	1.8 ± 0.3	1.9 ± 0.2	1.3 ± 0.1	1.2 ± 0.3
S, g kg^−1^	1.9 ± 0.1	2.0 ± 0.1	2.2 ± 0.1	1.6 ± 0.1	1.7 ± 0.1	1.7 ± 0.5	3.0 ± 0.4	2.1 ± 0.1
Sr, mg kg^−1^	43.5 ± 8.2	40.8 ± 2.3	64.6 ± 3.2	111.6 ± 30.2	39.0 ± 1.7	1.9 ± 0.1	39.0 ± 4.5	39.6 ± 12.5
Zn, mg kg^−1^	36.4 ± 3.1	43.5 ± 6.7	42.7 ± 3.7	26.9 ± 2.9	99.2 ± 25.3	67.2 ± 6.0	41.5 ± 0.9	31.1 ± 0.8

## Data Availability

Date is contained within the article and Appendix A.

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
