# Peer review of "Usefulness of Tree Species as Urban Health Indicators"

_plants, 2021, doi:10.3390/plants10122797_

Round 1
Reviewer 1 Report
The manuscript needs to be checked by an English peer. I started to sign the mistakes, then I gave up because there were too many. Latin names must be used throughout the text, especially in tables and figures.
I don't understand why the Authors chose to detect PM5 instead of PM2.5 as indicated by the international literature. This must be duly explained also citing references.
There are also some speculative sentences in the discussion not supported by the results or by the parameter measured
Author Response
Comments and Suggestions for Authors
The manuscript needs to be checked by an English peer. I started to sign the mistakes, then I gave up because there were too many. Latin names must be used throughout the text, especially in tables and figures.
ANSWER: We corrected the names of tree species to Latin names throughout the text, tables and figures.
I don't understand why the Authors chose to detect PM5 instead of PM2.5 as indicated by the international literature. This must be duly explained also citing references.
ANSWER: In case atmospheric fine particle content, PM10 and PM2.5 measurements have a reliable practice based on international patents. However, this is not obvious when we intend to measure the deposited dust, in our case on tree leaves, and we chose the washing and the filtering. The retention diameter of the used second type of filter paper of was 2-3 µm (Munktell 391, Ahlstrom) which allow to measure the PM5. The PM5 detection is also generally used to assess the level of air pollution (Agnihotri et al., Aerosol and Air Quality Research, 15: 58–71, 2015, Hoseinzadeh et al. Journal of Environmental Health and Sustainable Development, 5, 1035-1042., 2020, Massey et al., Building and Environment 47, 223-231, 2012). We completed our manuscript with a description of the importance of PM5 measuring (Line 30-35).
There are also some speculative sentences in the discussion not supported by the results or by the parameter measured.
ANSWER: We rewrote the Discussion section from Line 175-253.

Reviewer 2 Report
The abstract needs to be deeply revised: "elemental analysis", which one? "assessed the sensitivity", sensitivity to? how is defined the sensitivity and tolerance to air pollutants? "we demonstrated", how? "significant difference", unclear, which kind of difference? p value? Last sentence: to be deeply reformulated. "urban health", define this concept. Add the time period of the study.
References 1 & 2 seem not suitable: #2 is a paper in Ukraine during wildfire event, and #1 is about HAPs in soils. I found one ref (DOI: 10.1186/s12302-020-00450-2) stated that between 2000 and 2017, the annual PM2.5-related number of deaths decreased in line with a reduction of PM2.5 levels observed at urban air quality monitoring stations.
"in high quantities", to be quantified.
Ref 15 is about wind erosion in Hungary. In cities, PM2.5 have potentially the most significant effects on human health associated with respiratory and cardiovascular diseases and mortality, compared to other air pollutants (DOI: 10.1016/j.scitotenv.2013.01.077; DOI: 10.1016/S0140-6736(17)30505-6 ).
Move the section Materials and Methods as Section 2.
The section "Discussion" is too short and needs to be deeply developed and argued.
Section "Conclusions" - Second sentence, please correct or explain. You stated "silver maple" is not sensitive to the air pollution and "was moderately tolerant" just after.
Some typos through the manuscript e.g. (, & Tilia eurpaea in section 4.1 - To be cross-checked.
Author Response
Comments and Suggestions for Authors
The abstract needs to be deeply revised: "elemental analysis", which one? "assessed the sensitivity", sensitivity to? how is defined the sensitivity and tolerance to air pollutants? "we demonstrated", how? "significant difference", unclear, which kind of difference? p value? Last sentence: to be deeply reformulated. "urban health", define this concept. Add the time period of the study.
ANSWER: We corrected the Abstract from Line 13-22.
References 1 & 2 seem not suitable: #2 is a paper in Ukraine during wildfire event, and #1 is about HAPs in soils. I found one ref (DOI: 10.1186/s12302-020-00450-2) stated that between 2000 and 2017, the annual PM2.5-related number of deaths decreased in line with a reduction of PM2.5 levels observed at urban air quality monitoring stations.
ANSWER: Thank you so much for your recommendation. We corrected the references and we completed the Introduction section from Line 40-69.
"in high quantities", to be quantified.
ANSWER: We quantified it.
Ref 15 is about wind erosion in Hungary. In cities, PM2.5 have potentially the most significant effects on human health associated with respiratory and cardiovascular diseases and mortality, compared to other air pollutants (DOI: 10.1016/j.scitotenv.2013.01.077; DOI: 10.1016/S0140-6736(17)30505-6 ).
ANSWER: Thank you so much for your recommendation. We corrected the references.
Move the section Materials and Methods as Section 2.
ANSWER: The Materials and Methods is Section 4. It is the request of the Journal.
The section "Discussion" is too short and needs to be deeply developed and argued.
ANSWER: We rewrote and corrected the Discussion section from Line 175-253.
Section "Conclusions" - Second sentence, please correct or explain. You stated "silver maple" is not sensitive to the air pollution and "was moderately tolerant" just after.
ANSWER: We corrected it.
Some typos through the manuscript e.g. (, & Tilia eurpaea in section 4.1 - To be cross-checked.
ANSWER: Thank you so much your work. We checked the manuscript.

Reviewer 3 Report
The review concerned an article entitled: Usefulness of tree species as urban health indicators. The topic of air pollution is still a very important issue, especially in the urban areas described by the authors. In addition, there are many publications describing the ability of particular plant species to accumulate PM, while there are much fewer works on the usefulness of plants as bioindicators of particulate air pollution based on APTI index. As my native language is not English I will not check the manuscript for correct language.
Introduction - The introduction is too short and concise, it could be longer and more elaborated so that the information contained in it would give a more complete picture of the problem and a better understanding of the topic.
Line 28 – Only in large cities? Do the authors know that often in smaller towns (with a lot of single-family houses) air pollution is higher than in large agglomerations, as described by other authors?
Line 36 – There is no citation of this information. I haven’t seen any articles about the uptake of PM by the roots.? It is just the authors idea or facts ?
Line 41 – What is typical pollutant.?
Line 46.- What is the difference between dust and PM ? There any standards ?
Results - In my opinion, the authors should more often use Latin names of plants, not every reader is familiar with English-language nomenclature and may have trouble recognizing a given plant. Additionally, I think the authors should have broken down the results more. Most of the subchapter deals with correlations and significant differences, only at the end are the differences in results between the tested plants described.
Line 65-67- I understand that this is not the main topic of the publication, but 3 lines of text about PM accumulation is too little, especially for a generalized description. Its not acceptable.
Line 111-118- The description lacks specific differences, e. g. percentage ratios between the tested plants which hinders proper evaluation of the results.
Figure 1a and 1b – What is the dust content? It’s the amount of PM10?
Figure 2 and 3 – It should be shown and described statistical differences between plants.
Table 2. This table is not easy to read. There is no statistical differences between Elements.
Discussion - Looking at the amount of results and information, the authors have included too few authors with whom they are discussing. Information about the discussion does not appear until the end of the section.
Line 122-124. Authors does not measured the anatomy of the leaves. There is no data about it. This can be info can be in the introduction or with citation of specific articles. A lot of them is available: eg.
Popek R., Łukowski A., Karolewski P. 2017. Particulate matter accumulation – further differences between native Prunus padus and non-native P. serotina
Haynes A., Popek R., Boles M., Paton-Walsh C., Robinson S.A. 2019. Roadside moss turfs in South East Australia capture more particulate matter along an urban gradient than a common native tree species
Line 139 – There is very short discussion about the accumulation of PM. There is also a lot of articles about this topic. Author cannot concentrate only on one. In other publ. for eg. Amount of PM on the leaves of birch is very high like in this research: Łukowski A., Popek R., Karolewski P. 2020. Particulate matter on foliage of Betula pendula, Quercus robur, and Tilia cordata: Deposition and ecophysiology.
Line 137,143 - This paragraph have old data (more than 10 years). The data about PM are changing very often and there is a lot of new findings.
Line 157 – There is unneeded coma
Line 159 – There is lack of the authors of species name after every species in this sentence.
Line 162 – There is no information if the plants were planted in the center of the part or near the road. Its very important? There are great differences in accumulation in those places?
Line 162 – There is no info about the level of pollution on this area. Can authors can do the air pollution measurements?
Line 163- authors do not provide information about estimated age of the plants and their condition (presence of pests, symptoms of diseases).
Line 173. Why the sieve of 150 micormeters was used. The PM are as is known particles less than 100 micrometers.?
Line 176 and 178. Authors heave used filter papers whist diameter 5-8 and 2-3 micrometers. So the first on the filter stayed PM between 150-5 micromenetrs then on the other from 2-til 5. PM10 are all particles less than 10 micrometers, PM5 all of particles less than 5. Please explain what authors exactly have been measuring?
Line 230 – Why on the end the full names of the elements are used ?
Author Response
Comments and Suggestions for Authors
The review concerned an article entitled: Usefulness of tree species as urban health indicators. The topic of air pollution is still a very important issue, especially in the urban areas described by the authors. In addition, there are many publications describing the ability of particular plant species to accumulate PM, while there are much fewer works on the usefulness of plants as bioindicators of particulate air pollution based on APTI index. As my native language is not English I will not check the manuscript for correct language.
Introduction - The introduction is too short and concise, it could be longer and more elaborated so that the information contained in it would give a more complete picture of the problem and a better understanding of the topic.
ANSWER: Thank you so much your recommendation. We completed the Introduction section from Line 40-69.
Line 28 – Only in large cities? Do the authors know that often in smaller towns (with a lot of single-family houses) air pollution is higher than in large agglomerations, as described by other authors?
ANSWER: We agree with your comment, and we corrected the sentence.
Line 36 – There is no citation of this information. I haven’t seen any articles about the uptake of PM by the roots.? It is just the authors idea or facts ?
ANSWER: We corrected the sentence: the intensity of translocation through plant parts depends on the given metal and plant species.
Line 41 – What is typical pollutant.?
ANSWER: We completed the text from Line 44-45.
Line 46.- What is the difference between dust and PM ? There any standards ?
ANSWER: The atmospheric particulate matter (PM) consists of a complex mixture of substances and chemical compounds of natural and/or anthropogenic origin. Dust often referred to as particulate matter (PM). We completed the part of Introduction about PM and its type from Line 47-69.
Results - In my opinion, the authors should more often use Latin names of plants, not every reader is familiar with English-language nomenclature and may have trouble recognizing a given plant. Additionally, I think the authors should have broken down the results more. Most of the subchapter deals with correlations and significant differences, only at the end are the differences in results between the tested plants described.
ANSWER: We corrected the name of plants, and we use their Latin names. We summarized the results of dust and APTI to one subchapter.
Line 65-67- I understand that this is not the main topic of the publication, but 3 lines of text about PM accumulation is too little, especially for a generalized description. Its not acceptable.
ANSWER: Thank you so much your recommendation. We summarized the results of dust and APTI in a subsection.
Line 111-118- The description lacks specific differences, e. g. percentage ratios between the tested plants which hinders proper evaluation of the results.
Figure 1a and 1b – What is the dust content? It’s the amount of PM10?
ANSWER: We corrected it. The amount of PM10 is correct.
Figure 2 and 3 – It should be shown and described statistical differences between plants.
ANSWER: Thank you so much your recommendation. We indicated the statistical differences among species.
Table 2. This table is not easy to read. There is no statistical differences between Elements.
ANSWER: We agree with the reviewer, thus, we moved the Table to the Supplementary Materials section. The Table 2 includes the basic statistical data about canonical discriminant analysis. This demonstrates the validity of the statistical analysis; therefore necessary to include into the manuscript (Appendix is a good compromise). Fig 4. shows the results of Discriminant analysis. It is a multivariate method which is used to analyse multivariate samples of the elemental concentrations. In the studied case it reveals clearly that the species were really different based on the elemental concentrations of tree leaves.
Discussion - Looking at the amount of results and information, the authors have included too few authors with whom they are discussing. Information about the discussion does not appear until the end of the section.
ANSWER: We rewrote and corrected the Discussion section from Line 175-253.
Line 122-124. Authors does not measured the anatomy of the leaves. There is no data about it. This can be info can be in the introduction or with citation of specific articles. A lot of them is available: eg.
Popek R., Łukowski A., Karolewski P. 2017. Particulate matter accumulation – further differences between native Prunus padus and non-native P. serotina
Haynes A., Popek R., Boles M., Paton-Walsh C., Robinson S.A. 2019. Roadside moss turfs in South East Australia capture more particulate matter along an urban gradient than a common native tree species.
ANSWER: Thank you so much the recommended papers. We completed the Section of Discussion.
Line 139 – There is very short discussion about the accumulation of PM. There is also a lot of articles about this topic. Author cannot concentrate only on one. In other publ. for eg. Amount of PM on the leaves of birch is very high like in this research: Łukowski A., Popek R., Karolewski P. 2020. Particulate matter on foliage of Betula pendula, Quercus robur, and Tilia cordata: Deposition and ecophysiology.
ANSWER: We completed the Section of Discussion.
Line 137,143 - This paragraph have old data (more than 10 years). The data about PM are changing very often and there is a lot of new findings.
ANSWER: We completed this part of the Discussion.
Line 157 – There is unneeded coma
ANSER: We corrected it.
Line 159 – There is lack of the authors of species name after every species in this sentence.
ANSWER: We corrected it.
Line 162 – There is no information if the plants were planted in the center of the part or near the road. Its very important? There are great differences in accumulation in those places?
ANSWER: Plants were planted in the center of the studied area.
Line 162 – There is no info about the level of pollution on this area. Can authors can do the air pollution measurements?
ANSWER: We presented the results of PM 10 and PM2.5 over the time period 2000–2017.We completed the Materials and Methods section from Line 256-271.
Line 163- authors do not provide information about estimated age of the plants and their condition (presence of pests, symptoms of diseases).
ANSWER: We completed the section of Material and Methods section with the necessary information.
Line 173. Why the sieve of 150 micrometer was used? The PM are as is known particles less than 100 micrometer?
ANSWER: We agree with the reviewer’s comments that the diameter of PM is 100 micrometer. But first we used a sieve of 150 micrometer to remove the higher particles than 150 micrometer.
Line 176 and 178. Authors heave used filter papers whist diameter 5-8 and 2-3 micrometers. So the first on the filter stayed PM between 150-5 micromenetrs then on the other from 2-til 5. PM10 are all particles less than 10 micrometers, PM5 all of particles less than 5. Please explain what authors exactly have been measuring?
ANSWER: With the first filter paper (diameter 5-8µm) we can measure the PM10 which includes all the particles less than10 µm. With the second filter paper (diameter 2-3µm) we can measure the PM5 which includes all the particles less than 5 µm.
Line 230 – Why on the end the full names of the elements are used ?
ANSWER: We corrected it.

Reviewer 4 Report
The idea is not new, many papers regard the role of the tree in the urban air pollution. The authors should show more information on the heavy metal content, expecially showing a comparison on other European countries (for instance, see different papers by Capannesi et al. on the metals content in lichens) and stress the discussion in this contest. An English revision is necessary, in some points the discussion is quite difficult to follow. In Figure 4 the x- and y-axes must be reported for a good understanding of the plot. On the contrary, Table 1 should be simplified for evidencing the main results. Finally, give more details on the ICP-OES analysis
Author Response
ANSWER: Thank you so much recommended papers. We completed the Section of Discussion. Please find the inserted table which includes the instrumental conditions for ICP-OES for trace elemental analysis which is in the Supplementary Materials Section.
Fig 4. contents the results of Discriminant analysis which is a multivariate method which analyse multivariate sample of the elemental concentrations. This motivates the usage of discriminate analysis. In the studied case it reveals clearly that the species were really different based on the elemental concentrations of tree leaves.
|
Wavelengths/nm |
Al: 396.152 |
|
|
Ba: 455.403 |
|
|
Ca: 422.673 |
|
|
Cd: 226.502 |
|
|
Co: 228.615 |
|
|
Cr: 267.716 |
|
|
Cu: 324.754 |
|
|
Fe: 238.204 |
|
|
K: 766.491 |
|
|
Li: 670.783 |
|
|
Mg: 279.553 |
|
|
Mn: 257.610 |
|
|
Na: 589.592 |
|
|
Ni: 216.555 |
|
|
Pb: 220.353 |
|
|
Sr: 407.771 |
|
|
Zn: 213.857 |
|
Background correction |
fitted |
|
Number of replicates |
3 |
|
Pump speed (rpm) |
15 |
|
Rinse time (sec) |
30 |
|
Nebulizer flow (l/min) |
0.7 |
|
Analysis pump rate (rpm) |
15 |
|
Plasma mode |
dual |
|
RF power (kW) |
1.2 |
|
Read time (sec) |
5 |
|
Plasma flow (l/min) |
12.0 |
|
Sample introduction |
SPS 4 autosampler |
For sample introduction, double pass spray chamber and Meinhard type nebulizer was applied.

Round 2
Reviewer 1 Report
See comment in the manuscript file

Author Response
Comments and Suggestions for Authors
See comment in the manuscript file
Line 13: Use this taxonomy throughout the manuscript instead of Tilia europaea.
Answer: We corrected.
Line 15: „of the species” Don't need to write this.
Answer: We corrected.
Line 28: It's actually in some part of the world, while in Europe air pollution, though it's still a big issues, is dimishing https://www.eea.europa.eu/highlights/air-pollution-goes-down-as#:~:text=Although%20emissions%20of%20air%20pollutants,environmental%20health%20risk%20in%20Europe.
or
https://www.eea.europa.eu/publications/air-quality-in-europe-2021/health-impacts-of-air-pollution
Answer: We inserted this important information to the text.
Line 34: see also this https://www.mdpi.com/2071-1050/12/10/4247.
Answer: Thank you so much your recommendation. We inserted the paper.
Line 73: vary. Tolerance and sensitivity. So it's plural
Answer: We corrected.
Line 101: I frankly don't undestandand How did you assign the significance letters. How can Platanus not significantly different from Acer saccharinum and, on the opposite from all the other species?
Answer: We corrected.
Line 110: See comment above. I think you should clarify how you assigned the significance letteres
Answer: We corrected.
Line 157: space needed
Answer: We corrected.
Line 165: In this case you cite PM2,5, but you must explain why you compare PM5 to PM2,3 accumulation
Answer: We corrected.
Line 174: hairy or tomentose are more appriopriate terms than villous
Answer: We corrected.
Line 203: When you haven't mentioned the species before, you have to write the full name.
Answer: We corrected.
Line 207: When you haven't mentioned the species before, you have to write the full name.
Answer: We corrected.
Line 210: are
Answer: We corrected.

Reviewer 2 Report
All requested changes have been done.
Author Response
All requested changes have been done.
Answer: Thank you so much your work.
